# Kierkegaard, "the Public", and the Vices of Virtue-Signaling: The Dangers of Social Comparison

John Lippitt

The Institute for Ethics & Society, The University of Notre Dame Australia, 104 Broadway, P.O. Box 944, Broadway, NSW 2007, Australia; john.lippitt@nd.edu.au

**Abstract:** Concerns about the dangers of social comparison emerge in multiples places in Kierkegaard's authorship. I argue that these concerns—and his critique of the role of "the public"—take on a new relevance in the digital age. In this article, I focus on one area where concerns about the risks of social comparison are paramount: the contemporary debate about moral grandstanding or "virtue-signaling". Neil Levy and Evan Westra have recently attempted to defend virtue-signaling against Justin Tosi and Brandon Warmke's critique. I argue that these defences fail and that a consideration of epistemic bubbles and echo chambers is critical to seeing why. The over-confidence to which they give rise exacerbates certain vices with the potential to do moral, social and epistemic harm: I focus in particular on self-righteousness (complementing Kierkegaard's discussion of envy). I then argue that Kierkegaard's contrast between the religious category of the "single individual"—the genuine person of "character"—and the person who effectively appeals to the authority of some version of "the public" deepens our understanding of why we should reject defences of virtue-signaling. It helps us to distinguish between two kinds of virtue-signaler ("superficial enthusiasts" and "clear-eyed cynics"), both of whom contribute, in different ways, to the negative impacts of the vice of self-righteousness. Contrary to Levy's claim that virtue-signaling is virtuous, I conclude that typically it is closer to vice than to virtue.

**Keywords:** Kierkegaard; virtue; vice; social comparison; virtue-signaling; epistemic bubbles; echo chambers; self-righteousness

## 1. Introduction

One set of obstacles to the cultivation of genuine virtues are the various virtue-resembling masks in which vices sometimes hide themselves. Just as sinful pride can mask itself as proper self-respect, or the spirit of vengefulness pass itself off as the desire for justice, so recent years have seen the term "virtue-signaling" become ubiquitous. The term irritates many, who charge its users with intellectual laziness and complain about its over-use in the culture wars (Shariatmadari 2016; Leith 2020). But it is unsurprising that virtue-signaling (or "moral grandstanding") should attract condemnation.[1] These phenomena involve attempting to enhance one's moral reputation by easy means (especially on social media)—a concern with the appearance of virtue rather than its reality.[2] Like the examples of pride and vengefulness above, they too pass off as virtue something that is not. An interesting philosophical literature has recently emerged about virtue-signaling or moral grandstanding. Central to Justin Tosi and Brandon Warmke's landmark critique of grandstanding (Tosi and Warmke 2016, 2020a) is the claim that it interferes with the deliberative function of public moral discourse by replacing reason- and evidence-giving with social comparison. Kierkegaard too is deeply concerned about the dangers of social comparison, but his voice is absent from the above literature. In this article, I aim to show that Kierkegaard's concerns—and his ethical and religious critique of the role of "the public"—take on a new relevance in the polarised world of the digital age in which the discussion of virtue-signaling has its home. Tosi and Warmke's opponents have claimed

that virtue-signaling provides higher order evidence (Levy 2021) and is a potential aid to moral progress (Westra 2021). Levy even claims that virtue-signaling is itself virtuous. In what follows, I argue that these defences fail, and that reflection on certain vices (especially self-righteousness) in the light of Kierkegaard's critique of "the public" enables us more clearly to see why. A significant amount of recent work has considered Kierkegaard's relation to the virtue tradition and whether he can fairly be considered as a kind of "virtue ethicist".[3] It is usually agreed that if Kierkegaard is such, he is one of a specific kind, one who aims at the "upbuilding" of the whole person, focusing on their religious and not just moral development; whose approach to the virtues therefore incorporates specifically religious conceptions of faith, hope, love, gratitude, etc.; who denies that the cultivation of virtues is "meritorious", the development of such qualities being a gift of grace; and who (perhaps implicit in all of the above) is especially concerned with how virtues are transformed by Christian revelation and faith.[4] All honour to such scholarship, as Kierkegaard's pseudonym Johannes Climacus would say. But my concern here is different: to connect Kierkegaard for the first time with the philosophical debate about the qualities—virtuous or vicious—of virtue-signaling, and to show what his thought can contribute to this debate. I aim to show how Kierkegaard's view of social comparison and "the public" helps us to understand that virtue-signaling is typically closer to vice than to virtue.

My argument proceeds as follows. To do justice to that debate on its own terms, in Section 2, I analyse Tosi and Warmke's account of moral grandstanding and aspects of Neil Levy's and Evan Westra's respective replies defending—what they call—virtue-signaling.[5] Levy's higher-order evidence defence, I argue, overlooks the prevalence and dangers of epistemic bubbles and echo chambers and the illegitimately inflated epistemic self-confidence that they sponsor. The same problem haunts Westra's alternative defence of virtue-signaling, which also—by focusing on virtue-signaling's possible role in moral progress—overlooks its possible role in moral *regress*, thereby effectively begging the question.

In Section 3, I argue that the inflated epistemic self-confidence to which epistemic bubbles and echo chambers give rise exacerbate the risks of certain vices. I focus on a widespread but under-explored vice particularly pertinent to the digital age: self-righteousness. The biblical tradition has had more to say about this vice than secular philosophy. Drawing on a discussion of a paradigmatic case of self-righteousness from that tradition, the parable of the Pharisee and the tax collector in Luke 18—discussed by Kierkegaard in his Friday communion discourses (DCF 100-7/SKS 11, 263-9)[6]—I sketch some of its key features: its unforgiving nature, its tendency to abuse moral or religious conviction by making unwarranted claims to moral or religious certainty, its often-distorted sense of moral or religious values, and its avoidance of moral or religious complexity through selective ignorance and normatively slanted descriptions. A particular problem is the tendency to over-simplify complex moral or religious issues on which conscientious people may reasonably disagree, where the self-righteous make claims to certainty that outstrip the available justifications. But self-righteousness is not only a moral or religious problem. Connecting this issue with a recent discussion of intellectual arrogance and silencing (and their relation to the vice of servility), I outline some of the epistemic harms that can be laid at the door of self-righteousness: both to persons silenced and to the wider epistemic community.

In Section 4, I connect the above discussion explicitly to Kierkegaard's critique of "the public". I aim to show the importance of the "single individual" as the locus of "character", and that the dangers of "reflection" in Kierkegaard's pejorative sense, and of "leveling", are overlooked in the virtue-signaling debate. I further argue that two groups involved in virtue-signaling—what Westra calls "sincere advocates for change" and the less sincere virtue-signalers who follow them—are prone to different vices: self-righteousness on the one hand and the cluster of flaws that Kierkegaard calls lack of "character" on the other. However, these problems are connected: because self-righteousness can be a vice of groups as well as individuals (and one exacerbated by epistemic bubbles and echo chambers),

the impact of this vice can spread through virtue-signaling, in two ways that I sketch. Kierkegaard's analysis of "the public", applied to the virtue-signaling debate, enables us to distinguish between two kinds of virtue-signaler: the *superficial enthusiast* and *the clear-eyed cynic*. His critique underlines why we should be wary of the insufficiently critical reliance on social norms made by virtue-signaling's defenders. Furthermore, his analysis of the contrast between genuine "character" and the deficiencies of the "present age" offers new, additional reasons not to rest content with the virtue-signaler's lack of deep character. The insufficiently critical reliance on social norms typical of the virtue-signaler cannot replace the need for the kind of inner moral and religious integrity that Kierkegaard associates with the person of genuine "character".

## 2. Moral Grandstanding and Virtue-Signaling: For and Against

What is moral grandstanding?[7] For Tosi and Warmke, it has two key features: a "recognition desire" and a "grandstanding expression". The former amounts to wanting to impress others with one's moral qualities: either simply to "pass" as morally "respectable" with a given in-group or to appear as especially morally impressive (Tosi and Warmke 2016, p. 200).[8] The "grandstanding expression", which can be written or verbal, amounts to aiming to satisfy that desire via public moral discourse. A person grandstands "only if the Recognition Desire is a significant motivator for what she says" (Tosi and Warmke 2020a, p. 173). So grandstanding, to count as such, must involve the desire for recognition as morally respectable (in at least the more minimal sense), as well as being the expression of commitment to a moral or social norm. In short, it is a form of norm-signaling, the distinguishing feature of which is the "recognition desire".

Consider the charge that grandstanding interferes with a primary function of public moral discourse by substituting social comparison—which is not truth-sensitive—for the exchange of reasons, argument, and evidence. In response, Levy claims that although the practice has its risks and pathologies, it also has virtues that outweigh its vices. Virtue-signaling provides *higher-order* evidence to agents. Moreover, "signaling" is a primary, and valuable, function of public moral discourse, not a perversion thereof. While I have no objection to Levy's last claim, his general argument about "signaling" defends too broad a notion of virtue-signaling. To see this—and other, more fundamental, problems with his account—let's dig into the details.

As noted, Tosi and Warmke refer to two different varieties of grandstanding: seeking "moral respectability" in a minimal sense and gaining social status by showing oneself as "morally exceptional" (Tosi and Warmke 2016, p. 199; 2020a, p. 23). Yet in both cases, the grandstander is by definition concerned with *herself* rather than just the issues about which she claims to be concerned. The worry is that the recognition desire means that in signaling her (supposed) moral insight, she turns moral discourse into a "vanity project" (Grubbs et al. 2019).[9] Accordingly, Kierkegaard would likely judge the recognition desire as a manifestation of improper self-love.[10]

### 2.1. "Moral Respectability" and "Social Status"

At first glance, the quest for "moral respectability" in the more minimal sense might seem *too* minimal a claim: who could object to such "signaling"? However, what of issues on which a given in-group is committed to a particular view, taken to be "settled", which a broader community sees as far from settled? There can be circumstances in which showing oneself as "morally respectable" in a community where the matter in question has already been decided might be seen as a necessary career preservation strategy, regardless of the truth of the matter. This complicates Levy's claim that moral signals are "potentially costly" (since committing oneself to any moral position invites criticism if one falls short (Levy 2021, p. 9555)). Contrary to Levy's assumption that neutrality is the non-costly option, in such cases, *not* showing that one is on the side approved by the in-group is what is likely to prove costly.

Such cases may involve outright cynicism. But there is a more subtle possibility. In *The Power of the Powerless*, Vaclav Havel tells a memorable story of a greengrocer who (at the height of Czechoslovak Communism) displays a sign reading "Workers of the World, Unite!". He does so not out of any burning conviction but because refusal could lead to trouble, whereas compliance ensures "a relatively tranquil life" (Havel 2018, pp. 14–15). The slogan signals "a subliminal but very definite message. . . 'I know what I must do. I behave in the manner expected of me. I can be depended upon and am beyond reproach. I am obedient and therefore have the right to be left in peace'" (Havel 2018, p. 15).

Many quests for "moral respectability" on social media have this form ("I, too, hold the approved views, so leave me alone"). But as Havel astutely notes, few want to think this is all they are doing: if the greengrocer had been obliged to display a sign directly acknowledging an obedience rooted in fear, he would have been ashamed to do so. He must express his loyalty in a way that on the surface shows "disinterested conviction", allowing him to say, "What's wrong with the workers of the world uniting?", and thus "to conceal from himself the low foundations of his obedience", hiding his low motives from his conscience by masking them behind *ideology* (Havel 2018, p. 16). Thus, the quest for mere "moral respectability" is far murkier than it first appears. Naturally, the second variety of grandstanding—using moral talk to seek social status for being unusually morally impressive—sounds more objectionable than seeking *mere* "moral respectability", and it is within this territory that grandstanding and virtue-signaling are more likely to be used as pejorative terms. However, we should note that even the more minimal desire for respectability can be problematic. If I am right that much of what is posted on social media echoes the attitudes of Havel's greengrocer—if people feel pressured into signaling their support for "approved" views that are nevertheless morally contentious in order just to "pass" as morally respectable to a given in-group—then that is a matter of great concern for the quality of public moral discourse.[11] I shall return to this in discussing epistemic injustice and silencing in Section 3.

### 2.2. Grandstanding: Forms and Effects

What are the main forms of moral grandstanding? Tosi and Warmke outline five:

(1) *Piling on*. Reiterating something that has already been said in order "to get in on the action" (Tosi and Warmke 2016, p. 203), registering one's view in the public square, even if doing so adds nothing new. They surmise that people engage in such behaviours because they want to be perceived (and to perceive themselves) favourably: as with Havel's greengrocer, it's a signal that one "fits".

(2) *Ramping up.* One person's demand for the public censure of a wrongdoing senator leads another to suggest seeking their removal from office and still another to suggest pursuing criminal charges. This "moral arms race" (Tosi and Warmke 2016, p. 205) is unsurprising if the grandstander is motivated to show herself as more morally respectable than others. This can lead the original complainant to recalibrate their position to something still more extreme to maintain the image they wish to present (Tosi and Warmke 2016, p. 206).

(3) *Trumping up.* Moralising a non-moral problem. Trumping up seeks to demonstrate that the grandstander has a keener moral sense than most, noticing as "problematic" what most people have missed. But the "problem" identified is in fact not morally objectionable.

(4) *Excessive outrage* (or other *strong emotions*). Emotion displays are often taken as proxies for strength of moral conviction. So, they can again be used strategically to communicate the grandstander's heightened moral convictions relative to others. While emotional reactions to immorality are often fitting, the grandstander's outrage is disproportionate to the seriousness of the issue.[12]

(5) Finally, high-handed *dismissiveness* of alternative views or *claims of the self-evident truth* of positions held. In other words, it is taken to be the case that any decent, right-thinking person would see things as the grandstander does, such that their

views need not be defended. Any doubts or suggestions by others that matters may be more complex can be declaimed as showing a deficiency in either the other's moral sensitivity or their commitment to morality itself (Tosi and Warmke 2016, p. 208).

In the light of this fifth form, I would qualify the third. For Tosi and Warmke, to trump up, one must "get something wrong" (Tosi and Warmke 2020a, p. 55). I suggest a broader account: one form that trumping up can take is insisting there is a moral problem where that claim is not so much *definitively wrong* as *controversial*, where the truth of the moral claim assumed is no more obvious than the view rejected as beyond the pale. The problem here is acting, as noted earlier, as if a matter of genuine moral controversy on which conscientious people disagree has already been "settled". To defend this position, some of the other features (*ramping up*, *excessive outrage*, and—especially—*claims of self-evident truth*) are typically displayed alongside *trumping up*. To anticipate Section 3, this is an arena in which the vice of self-righteousness often reveals itself. And to anticipate Section 4, it is an arena in which an unwarranted deference to some version of "the public" often plays out.

Tosi and Warmke consider three negative effects of grandstanding: *moral cynicism*, *outrage exhaustion*, and *group polarisation*.[13] By moral cynicism, they mean a form of "skepticism and disillusionment about the sincerity of people's contributions to moral discourse" (Tosi and Warmke 2016, p. 210). This is a kind of cynicism spiral: if the—cynical—motivation for grandstanding is to *appear* a certain way, and others come to see this as a common motive for making moral claims, they may naturally come to think that public moral discourse has become simply about *showing* oneself to be on the "right" side, regardless of sincerity. The currency of moral talk is thus devalued (Tosi and Warmke 2020a, p. 80).

Grandstanding's tendency towards excessive outrage leads Tosi and Warmke to make a two-fold prediction. First, that it will become harder to judge when outrage is "a reliable signal of injustice" (the crying wolf problem: if someone treats the "cultural appropriation" of food or costumes with the same outrage as police brutality, why suppose their outrage "tracks anything important"? (Tosi and Warmke 2020a, p. 84)). Second, that it will become harder to muster moral outrage when appropriate: hence *outrage exhaustion*.

*Group polarisation* is exacerbated because ramping and trumping up encourage a shift towards more "extreme" viewpoints. Sometimes, this is a deliberate attempt to shift the Overton Window: the range of ideas or policies that the public is willing to consider. But Tosi and Warmke focus elsewhere: that this shift towards more extreme views is a likely consequence of the fact of people engaging in social comparison to preserve their self-conception and reputation with the in-group. A further consequence is that outsiders come to be alienated from such moral discourse, viewing it as consisting mainly of "extreme and implausible claims" (Tosi and Warmke 2016, p. 212).

### 2.3. Levy's Defence of Virtue-Signaling

Levy is more sanguine about the resulting risks to the devaluation of moral discourse, claiming that "every practice has its pathologies" and that to judge "whether a practice should be condemned as a whole", we also need to analyse benefits as well as costs and whether they can be obtained by a "less costly" route (Levy 2021, pp. 9547–8). The preliminary reasons he takes his paper to sketch suggest that such an assessment would vindicate virtue-signaling.

Levy claims that virtue-signaling has two important functions. First, he argues that far from undermining the deliberative function of public moral discourse, virtue-signaling supports it. It does so by offering *higher-order evidence*: evidence about "the reliability of the processes that generate belief" (Levy 2021, p. 9548). How so? By conveying *confidence* and by the *numbers* of people who hold such a belief. Most of what we know is through testimony, and confidence is a useful heuristic in assessing testimony. The number of people supporting a view is a similarly useful heuristic: the more people I know to disagree with me, the harder it is to dismiss them by "citing my better track record, intelligence, lack of bias, and so on" (Levy 2021, p. 9549). Second, Levy claims that another function of

deliberative moral discourse is to help solve coordination problems. As social creatures, we need to share information and thus coordinate behaviour. But in a complex society, free-rider risks increase, and here is where virtue-signaling allegedly helps: by signaling to others that we are trustworthy. Virtue-signaling helps us to delineate who are reliable moral co-operators.[14]

I have four objections to Levy's approach. First, his description of "the practice as a whole" is misleading, as Tosi and Warmke are not objecting to *any* kind of "signaling our commitment to norms" (Levy 2021, op. cit., p. 9545, cf. p. 9554). As Jesse Hill and James Fanciullo have argued, whereas virtue-signaling entails norm-signaling, the converse is not true. One can—and many do—signal one's commitment to a norm without the "recognition desire"—the desire to gain credit by being seen as especially morally impressive or at least morally respectable in a more minimal sense—which, they rightly suggest, captures our "folk" concept of virtue-signaling. Whereas norm-signaling is indeed an important part of moral communication, this additional feature specific to virtue-signaling is not needed (Hill and Fanciullo 2023).[15]

Second, Tosi and Warmke's position does not require the view that "extreme" views are more likely to be false (Levy 2021, pp. 9550–1).[16] At its strongest, their objection is that—if their suggestions are correct—grandstanding *by its nature* (trumping up and especially ramping up) encourages a direction of travel towards greater "radicalisation". That is still a legitimate concern even if, on a specific matter, the truth lies beyond the current range of the Overton Window: there is still no reason to suppose that this is *in general* a desirable direction of travel.

Third, Levy's defence sets too low a bar for virtue-signaling to clear. Contrary to his claim that virtue-signaling is "virtuous", one obvious contrast between virtue-signalers and the genuinely virtuous is that on many dominant accounts of the virtues, the virtuous person does not think of virtues as means to acquire goods external to the virtues.[17] It is an important point against virtue-signalers that their motives—chiefly, their own moral reputation with the in-group—are, from this point of view, suspect.

This brings us, fourth, to the problems with Levy's defence that I will discuss in most detail, both of which pertain to the higher-order evidence claim. His focus on both *confidence* and the *weight of numbers* is problematic. Levy does not respond to the *outrage exhaustion* charge, except insofar as he suggests that moral outrage may be a clue to the degree of confidence people have in their moral judgements. He suggests that *polarisation* might come about through "rational agents optimally taking into account both the confidence with which testimony is offered and the number of agents who share an opinion" (Levy 2021, p. 9551).

However, much depends on the examples considered. On *confidence*, Levy's example is of two different answers to the question of where the railway station is—one confident and detailed, one tentative—which naturally favours the former's greater reliability. But such an example is a poor guide to the controversial moral territory on which the debate about virtue-signaling has its natural home. It may be true that people typically use what Levy calls a "confidence heuristic" in assessing testimony, but it does not follow that they are reasonable to do so in non-trivial examples. One routinely encounters people who express, with great confidence, views on topics about which they are ignorant. Grandstanders may do so with great confidence. But to rely in this way on confidence in domains of moral controversy is in practice already to have rejected the possibility that a deliberate—but contentious—attempt to shift the Overton Window is underway. There is no reason to suppose that the person who is part of this attempt can be relied upon any more in virtue of their doing so *confidently*—an additional reason for which we shall see next.[18]

### 2.4. Epistemic Bubbles and Echo Chambers

What of Levy's "*weight of numbers*" argument? To anticipate our later discussion, we might see this as the defence of a view that Kierkegaard roundly rejects: that "truth is where the crowd is" (PV 106/SKS 16, 86), Kierkegaard here defining the crowd as "the numerical"

(PV 107/SKS 16, 88). Levy's example here is again non-moral (in dividing a restaurant bill, the more competent reckoners who reach the same opposing result as me, the more reason I have to doubt my calculations). So far, so good. But what Levy underplays—a point also pertaining to the reliability of any "confidence heuristic", as we shall see—is the threat of various "epistemic bubbles" and "echo chambers". A powerful case has been made for the dangers of these phenomena, perhaps most significantly by C. Thi Nguyen. An epistemic bubble is "a social epistemic structure which has inadequate coverage through a process of exclusion by omission" (Nguyen 2020, p. 143). Simply by our tendency to obtain our news via social media, and to associate online with like-minded friends, we can—innocently enough—omit many sources of information and opinion that could properly influence our views on significant matters. These omissions mean we lack "coverage reliability"; we view the world through "a narrowed and self-reinforcing epistemic filter" (Nguyen 2020, p. 142). Epistemic bubbles provide us with "bootstrapped corroboration": we encounter agreement more often than would otherwise be the case, and so, our epistemic self-confidence is over-inflated (Nguyen 2020, pp. 143–4).[19]

Levy acknowledges that if the additional numbers are "non-discriminating reflectors of a single individual" (Levy 2021, p. 9549n3)—say unreflective followers of an online guru—then they add nothing. But he rejects this worry on two grounds. First, even if some agents are non-discriminating with respect to specific opinions, their regarding someone as a guru "is good evidence that they take them to be reliable in general" (Levy 2021, p. 9549n3). But the question at issue is whether they are *reasonable* in reaching this conclusion, and no case has been made for that. Second, there is "extensive evidence that agents are rarely genuinely non-discriminating" (Levy 2021, p. 9549n3): even small children filter testimony for plausibility to some degree. But in practice, agents do not have to be genuinely non-discriminating—just gullible. Which we all are, at least sometimes. The worry about epistemic bubbles cannot be dismissed so easily. Shortly, I'll qualify even the gullibility concession.

An echo chamber is far worse than an epistemic bubble. In Nguyen's usage, the former is a social epistemic structure in which a distinction between the in-group and out-groups is crucial. Far from mere accidental exclusion by omission, echo chambers actively and systematically isolate their members from outside epistemic sources (as in some accounts of cult indoctrination). Key to this is the manipulation of trust. An echo chamber is "an epistemic community which creates a significant disparity in trust between members and non-members", achieved by excluding non-members through "epistemic discrediting" (as well as by ramping up the epistemic credentials of members) (Nguyen 2020, p. 146). Further, general agreement with some core set of beliefs—including beliefs that support this trust disparity—is a prerequisite for membership. A certain kind of "closed" religious community would be an example.

It can be difficult to know whether you are in an echo chamber. While an epistemic bubble may be pricked with comparative ease (you stumble upon a radically different view, persuasively argued for), the dice of an echo chamber have already been loaded. As you have already been primed to distrust external sources of information as emanating from inherently untrustworthy out-groups, the new view has been discredited in advance. Not just isolated, you have been "actively alienated" (Nguyen 2020, p. 147) from such sources. Echo chambers—being excellent ways to bolster power—are often set up for this purpose. Escaping an echo chamber is thus much harder than escaping an epistemic bubble: it might require a lengthy, radical reboot of one's entire belief system.[20]

As noted, this requires us to qualify the above gullibility concession. Nguyen rightly notes that the gullibility charge is too easily made against echo chamber victims. Given social epistemology's view as to how dependent we are upon trust in others (recall that Levy accepts the idea that most knowledge relies upon testimony), to treat echo chamber victims as "mere sheep" is to rely on "an unreasonable expectation for radical epistemic authority" (Nguyen 2020, p. 142). Echo chambers can transform epistemically virtuous activity by an individual—being alert and engaged, vigorously and earnestly checking

sources—into a collective epistemic vice (because checking will, within that structure, throw up apparent reasons to distrust those sources). It is not these individual practices that are vicious but the social epistemic system in which the echo chamber victim has been raised (Nguyen 2020, p. 155). If alternative views are actively discredited by trust manipulation, then activities that are ordinarily epistemically virtuous can lead to the biases of the echo chamber being reinforced.[21]

This introduces a further problem, concerning biased information *processing* as opposed to biased information *exposure*. In the former, one conforms one's assessment of the available information to an end "extrinsic to accuracy" (Avnur 2020, p. 580), engaging in "motivated reasoning", such as subjecting evidence for claims *opposed* to one's interests or beliefs to more substantial critical scrutiny than those *favouring* them (Avnur 2020, p. 585). This worry is supported by sociological research, according to which *credulity is often an aspect of partisanship*: we are more likely to believe "our" side and disbelieve—or at least demand higher standards of proof from—the opposition (Campbell and Manning 2018, pp. 114–5).

In short, the prevalence of online epistemic bubbles and echo chambers is a serious obstacle to Levy's trust in both confidence and the weight of numbers as measures of the reliability of a belief-forming method. This worry is underlined if, as some evidence suggests, people often enter echo chambers to attain the sense of belonging to an in-group (Nguyen 2020, p. 154). While this prevalence may only be a contingent factor, it gives us reason to be far more pessimistic than Levy about the practical utility of virtue-signaling.

A second defence of virtue-signaling also pays insufficient regard to the problem of epistemic bubbles and echo chambers.[22] Westra's response to Tosi and Warmke is to argue that many social practices obstructing "moral progress" are motivated not by moral beliefs but by a strong desire to conform to local norms (Westra 2021, p. 159). So, to get people to abandon these practices, it is neither necessary nor sufficient to change their moral beliefs: rather, you need to get them to change the social beliefs that underpin what they expect about what their peers think and do. Virtue-signaling can aid moral progress by providing "a vector for information about social norms" (Westra 2021, p. 159). Westra draws on Cristina Bicchieri's account of a social norm as a rule of behaviour that individuals prefer to conform to on condition that they have the *empirical* expectation that "most people in their reference network conform to it" and the *normative* expectation that most people in that network believe that they *ought* to do so (Bicchieri 2017, p. 35; cited in Westra 2021, p. 160). Virtue-signalers tell their audience what they think should be impermissible or shameful (albeit primarily to appear admirable themselves). For Westra, this is valuable information for someone in the audience wishing to learn about the social norms of an in-group (Westra 2021, p. 164).

The problem is that descriptions such as Bicchieri's will work for epistemic bubbles just as they do for any other kind of "reference network".[23] So, the social norm revealed may be one that only exists *within* an epistemic bubble. Like Levy, Westra claims that "piling on" gives an audience member evidence about what *many* people believe ought to be the case (Westra 2021, p. 164; Levy 2021, p. 9554). But if epistemic bubbles are prevalent, this means that the audience member will find it difficult to distinguish between what a sufficiently sizable group (let alone "society at large"—or "the public") believes and what a small minority—perhaps a well-organised minority aiming to influence public opinion—believes. The corroboration is "bootstrapped". Even worse, if the in-group is not just an epistemic bubble but an echo chamber, then alternative views to this minority opinion will already have been silenced by discrediting, and so, the audience member will likely be unable to escape obtaining misleading information about the scale of the social norm.

This casts doubt on the significance of Westra's claim that virtue-signaling is potentially positive in shifting social norms (Westra 2021, p. 157). While it *can* have such a benign effect, there are no grounds for taking this as the default.[24] Westra describes the potential

utility of virtue-signaling "as a tool for positive norm change" ([Westra 2021](#), p. 164) as a three-step process:

> first, a group of sincere advocates for change seed a new, positive normative standard into the public discourse; second, virtue signalers eager to appear "on the side of the angels" broadcast this new standard to a broader audience through a mix of positive avowals and public shaming; third, a much larger population treats the behavior of these virtue signalers as evidence that they should change their social expectations, and become motivated to conform to the new norm. ([Westra 2021](#), p. 165)

Westra's example concerns climate change and the shift in attitudes from frequent flying as a hallmark of a glamorous lifestyle to something somewhat shameful ([Westra 2021](#), pp. 165–6). But the problem is that precisely the same structure works for *negative* norm change. Only by begging the question can we assume that moral *change* amounts to moral *progress*. So, even if we were to bracket objections about the dubiously self-interested motives of the virtue-signaler, it still seems a relatively weak defence of the practice that it *can* lead to desirable outcomes, provided the moral change is "positive", when *precisely the same process* can spread *negative* moral changes. One way that the latter can happen is through the silencing of reasonable opposition to the change (see further in Section [3](#)). Moreover, Westra's essentially consequentialist argument overlooks the life-damaging nastiness of many forms of "public shaming" (cf. [Ronson 2015](#)), which—it could reasonably be claimed—are morally indefensible even despite possible good consequences arising from a particular norm change.

### 3. Grandstanding's Vices: Self-Righteousness, Servility, and Epistemic Injustice

In this section, I propose that the over-confidence to which epistemic bubbles and echo chambers give rise exacerbates certain vices that can do moral, social, and epistemic damage. I focus on *self-righteousness*, a vice to which most of us are prone and yet that has received little philosophical attention. To show what is wrong with self-righteousness, I shall start from a paradigmatic religious case discussed by Kierkegaard, sketching some key features of this vice insofar as they pertain to grandstanding, before exploring how it can be *epistemically* harmful, as well as morally and socially objectionable.

The New Testament parable of the Pharisee and the tax collector targets an audience "confident of their own righteousness" who look down on others. The Pharisee prays: "God, I thank you that I am not like other people—robbers, evildoers, adulterers—or even like this tax-collector. I fast twice a week and give a tenth of all I get." By contrast, the tax collector simply beats his breast and says, "God, have mercy on me, a sinner." Jesus' message is that it is the usually reviled tax collector—the collaborator with the Roman forces—rather than the Pharisee who "went home justified before God. For all those who exalt themselves will be humbled, and those who humble themselves will be exalted" (Luke 18: 9–14, New International Version).

The target of critique here is someone with a sense of himself as normatively superior to others and who thereby assumes himself to have the standing to judge those others negatively. This Pharisee's self-righteous judgementalism is rooted in an apparent lack of awareness of his own fallibility, a blindness to his own sin and need for repentance. This is one form of the vice of pride. His self-righteousness is also a form of hypocrisy: he stands in judgement on the moral failings of others, while overlooking his own. In contrast, the tax collector focuses on his *own* sin, not how he compares. Acutely aware of his own imperfections, he therefore refrains from rushing to judge others. The Pharisee of the parable demonstrates a key feature of self-righteousness: paying insufficient attention to one's own wrongs and flaws blinds one to the need for a more generous and forgiving view of others. He fails to see that those on whom he stands in judgement have features that do not justify judging them simply as "sinners" or "wrongdoers", still less as moral monsters, beyond redemption. His unforgiving view of those on whom he looks down lacks *generosity of spirit*.

Note how central here is what Kierkegaard calls the spirit of *comparison*. In his Friday communion discourse on Luke 18: 13, he notes that this Pharisee uses the "fraudulent. . . criterion of human comparison" that occurs whenever "there is anyone between you and God" (DCF 102/SKS 11, 265). He has "with" him those to whom he compares himself (such as the tax collector), whereas the latter stands "far off", distancing himself from others and focusing simply on himself before God.[25] Self-righteousness, like envy (more of which is in Section 4), is one of the vices of pride in which the comparative aspect is most pronounced. Robert C. Roberts suggests that part of this vice involves an otherwise admirable desire for purity being turned into a moral stain and a defilement of one's religious character by caring more about one's *superiority to others* in some respect (Roberts 2022, p. 305). Kierkegaard has an interesting twist on this: the risk of being judgemental *of the Pharisee*, such that one effectively replaces his "I thank God I am not like this tax collector" with "I thank God I am not like this Pharisee" (DCF 101/SKS 11, 264). Accordingly, most of his discourse focuses on what is admirable about the tax collector, rather than in condemning the Pharisee. This offers an important warning: the risk of being self-righteous in judging others as self-righteous!

How does all this connect with the judgementalism of online grandstanding? First, mercilessness, or the unwillingness to forgive, are common features of pile-ons and other forms of social media take-downs. This world of finding guilty and seeking to shame is often motivated, I suggest, by self-righteousness. Having pitched one's tent on the moral or religious high ground, the inability to see oneself as a flawed creature in need of forgiveness is a key element in the unwillingness to extend forgiveness to others.

Second, extrapolating from this paradigmatic religious example of the Pharisee, I will sketch three further features of self-righteousness relevant to grandstanding. Blindness to one's own flaws and a concern with one's superiority to others give rise to three tendencies of the self-righteous person: to make *unwarranted claims to judgemental certainty* (rooted in a lack of awareness of the defeasibility of their own moral or religious judgements), to manifest a *distorted sense of moral or religious values*, and to issue *normatively slanted descriptions*.

First, self-righteousness often involves the degree of confidence expressed outstripping the justification that can be offered for a view. This often occurs when a complex moral or theological issue on which conscientious, well-informed people may reasonably disagree is treated as if it had been settled. The expression of convictions, or the pursuit of protest, cannot afford to lose sight of values of enquiry, such as open-mindedness to alternative views and responsiveness to reasons rather than postures. If Westra's "sincere advocates for change" fall into this category, then we should hardly celebrate their under-warranted moral judgements being passed on by less self-righteous virtue-signalers who want to show that they "fit". If his three-stage process is accurate, then unless we beg the question in favour of the moral correctness of the "sincere advocates for change", it gives us more worrying results than he recognises. In a case where the original protestors are prone to the limitations of the moral vision characteristic of the self-righteous person (such as making unwarranted claims to moral or religious certainty), virtue-signalers need not *themselves* be self-righteous to spread this vice's harms. Morally insincere virtue-signalers can transmit the dubious, perhaps false, judgements made by the influential original protestors, spreading their malign influence. We shall draw on Kierkegaard to further explore the dynamics of this in Section 4.

To flesh out this claim, consider Amélie Rorty's discussion of self-righteousness as one of several "abuses of morality".[26] Rorty's "self-righteousness" resembles Tosi and Warmke's "claim to self-evidence": abusing conviction "by *treating it as a claim to judgemental certainty*" (Rorty 2012, p. 8, my emphasis). This incorporates one way in which moral values are distorted, bringing us to our second tendency: the self-righteous person treats the function of morality as if its point was to provide judgement—"to issue in summary sentencing" (Rorty 2012, p. 8), contrary to Jesus' injunction against judgementalism (Matthew 7: 1–3). Tending towards an "us and them" attitude (and note the centrality of

*comparison* to such a view), his moral worldview typically lacks nuance or complexity, forgoing the hard work of genuinely understanding others that morality demands. He treats the heuristics of morality as definitive conclusions, claiming for himself "the right to identify and judge deviation, using formulae of exhortation without qualification, without context or interpretation" (Rorty 2012, p. 9). Accordingly, he tends to use moral discourse to truncate discussion and enquiry.

This connects with a related risk: that a focus on *à la mode* moral issues—such as, we shall see, those that Kierkegaard's "the public" is said to care about—can drown out moral issues of equal or greater concern (perhaps owing to availability bias), leading to another way in which moral values are distorted.

*How* does the self-righteous person or group express this distortion of values? This brings us to the third tendency. One method is by claiming the moral high ground by focusing on what Rorty calls a "selectively charged, normatively slanted description" (Rorty 2012, p. 10). This need not involve deliberately misrepresenting evidence. Sometimes, it simply involves presenting people under one description rather than another: she is a "Bible basher" rather than a committed Christian; he is a member of the "liberal metropolitan elite" rather than a highly educated professional; she is a "TERF" rather than a gender-critical feminist; he is a "cultural Marxist" rather than a political progressive. Such descriptions—like the "robbers, evildoers, adulterers" of Luke's parable—come to carry "value-laden, action-guiding implications" (Rorty 2012, p. 10). Through subtle and often self-deceptive means, the self-righteous (like Havel's greengrocer) remain "selectively ignorant" (Rorty 2012, p. 10), disguising from themselves what they are doing.

These phenomena are not just morally and socially objectionable. Self-righteousness is also *epistemically* objectionable, insofar as it perpetuates specifically epistemic injustices. How so?

I propose that self-righteousness, as sketched above, perpetuates "testimonial injustice". In one of Miranda Fricker's paradigm cases thereof, Tom Robinson's trial testimony (in *To Kill a Mockingbird*) is dismissed because of the jury's assumption that "all Negroes lie" (Fricker 2007, p. 25). To a prejudiced jury, Robinson has a "credibility deficit" (Fricker 2007, p. 28). For Fricker, he suffers testimonial injustice as follows: although his report conveys knowledge (he is in fact innocent), his audience does not accept this report. This non-acceptance is due to a distorting prejudice in the jury arising from his social identity and the relationship of social power between Tom and his audience. Moreover, prejudices relating to social identities—including religious ones—often operate in terms of "images" rather than beliefs, images that "can operate beneath the radar of our ordinary doxastic self-scrutiny" (Fricker 2007, p. 40).

The self-righteous person is disposed to flaws in important ways akin to those of the prejudiced jury. Having staked their claim to judgemental certainty, the self-righteous person effectively rejects the possibility that the testifier might have knowledge to bring to bear. The overly simplistic "normatively slanted description" ensures that the distorting prejudice is in place and based on social identity (taking this notion to extend beyond man or woman, black or white, gay or straight (Fricker 2007, pp. 14–17) to descriptions like "Bible basher", "metropolitan elite", "TERF", or "cultural Marxist"). Social power plays out in complex ways, and a danger of epistemic bubbles and echo chambers is that we may fail to see some of the ways in which power shifts over time and context—and thus of which groups may be systematically disadvantaged in which contexts.

We can therefore see that self-righteousness has the capacity to inflict significant epistemic harms on those over whom the self-righteous stand in judgement—especially if the self-righteous have the weight of in-group numbers on their side. Being judged in this way can lead, first, to being silenced and a consequent erosion of self-confidence, which over time can lead to what Alessandra Tanesini calls "servility":

> The silenced individuals will soon learn that it is less risky to share the views of those who are capable of silencing them. These individuals may bite their tongues unless what they think coincides with powerful views. Over time, one



may expect that because of cognitive dissonance such individuals may stop biting their tongues and simply defer to the opinions of others. When they do so, they have become servile. (Tanesini 2016, p. 90)[27]

More than one all-too-human vice may motivate the initial tongue-biting. Ethical or spiritual laziness (perhaps motivated by the appeal of a quiet, non-confrontational life) and timidity (the lack of moral courage) are both possible motives. However, as Sanford Goldberg has argued, the best way to explain the development of servility is not in terms of the desire to erase cognitive dissonance (Goldberg 2016).[28] One detail of Goldberg's alternative account is of particular interest in our response to Levy. Goldberg focuses on how in many cultures and contexts, silence is often interpreted as assent (or at least acquiescence) to another's assertion. So, when one is silenced by the intellectually arrogant or self-righteous in such a context, that silence will often be interpreted as having no reply available, so "*the very silence of those who are the victims of . . . oppression and subordination is itself standardly interpreted as further evidence for the warrantedness of the way these victims are being treated*" (Goldberg 2016, p. 96, emphasis in original). In other words, one's own silence contributes to one's continued disadvantage.

Particularly interesting here is the role of higher-order evidence (a key plank in Levy's argument, as we saw in Section 2). In contexts where silence is taken to mean assent, the silenced person is likely to interpret the silence of others as evidence that *they* assent to the speaker's assertion (Goldberg 2016, p. 105). In other words, this assumption thus provides plausible higher-order evidence that others agree with the self-righteous speaker. This higher-order evidence can lead the silenced person to doubt their own judgement that the speaker's assertions are in fact dubious. Moreover—and this is the key point—Levy's focus on the weight of numbers exacerbates the problem. In a large group, if none challenge the self-righteous speaker's assertions (perhaps owing to their power or charisma), the silenced person has greater epistemic grounds to doubt their own judgements. But power and charisma do not necessarily correlate with truthfulness or insight. Hence the problem:

> The road that eventuates in a loss of self-confidence and a deference to others' opinions can start off through the subject's appreciation of the evidence itself—evidence which, unbeknownst to her, is highly misleading (as the other participants might be silent, as she is, out of motives *other than acceptance*). (Goldberg 2016, p. 105)

What this adds to our argument is that in common situations of silencing, such misleading higher-order evidence can shift a person from mere silence (perhaps arising from the pressure felt to self-censor) to self-doubt to servility. If a person is in some important way disadvantaged, and their silence is interpreted as justification for the perpetuation of their disadvantage, then they are the victim of a distinct kind of epistemic injustice (Goldberg 2016, p. 109). But—even worse—such contexts lead to avoidable ignorance as the *broader epistemic community* is deprived of sources of potential knowledge and justified belief (Goldberg 2016, p. 109). Not only the silenced party but also the broader epistemic community has been damaged.

In line with the most common readings of the parable of the Pharisee and the tax collector, the limited philosophical literature on self-righteousness (cf. Hill 1979; Bicknell 2010; Rorty 2012; Zuckert 2021) tends to discuss it as a vice of individuals. But it should already be evident from the discussion above that self-righteousness can also be a *collective* vice: a vice possessed by groups. Bear this in mind as we focus our attention more squarely on what Kierkegaard adds to the conversation.

## 4. Kierkegaard, "The Public", and the Problems of Leveling

As we have seen, Westra notes the strong desire to conform to local norms as a common human motivator. For Kierkegaard, however, there is something intrinsically dubious about such an uncritical relation to local norms. One of Kierkegaard's key worries about mass society—the collectives he variously labels "the crowd" and "the public" and

about social comparison more generally—is that to take such a yardstick is to normalise an insufficiently critical process of adopting social norms. In a nutshell, to appeal to the authority of "the public" is to fail to be "the single individual". Since the "single individual" is for Kierkegaard a specifically religious category, this is at heart a religious failing. Several overlapping features are important in understanding this: Kierkegaard's general concerns about social comparison, the connection between being a "single individual" and the virtues, the danger of a "phantom" called "the public" as the instrument through which "leveling" takes place, and, finally, the link between "shrewdness", the use of "the public" as a yardstick for one's norms, and virtue-signaling. By exploring each of these features in turn, we shall see how Kierkegaard's critique gives us additional reasons to reject defences of virtue-signaling, as well as suggesting the need to distinguish two distinct *kinds* of virtue-signaler.

### 4.1. Kierkegaard's Worries about Social Comparison

> If I were to imagine a human being who was brought up in such a manner and lived out his life in such a manner that he never got any impression of himself but always lived by adaptation and comparison—this would be an example of dishonesty. And this is precisely the state of affairs in modern times. (JP 1, 654/SKS 27, 417)

Kierkegaard is deeply troubled by social comparison: the human tendency continually to evaluate one's identity and norms in relation to those around one (Kaftanski 2021, pp. 204–5). He associates this with ethical-religious laziness as well as dishonesty: "As soon as men become indolent and seek indulgence, they promptly escape into sociality, where the standard is relative, comparison with others". Far from being a "higher perfection", such social comparison is "retrogression" (JP 2, 2010/SKS 21, 128, NB7: 97). In a rich discussion of social comparison in the context of *mimesis*, Wojciech T. Kaftanski notes Kierkegaard's observation that we tend to search for situations in which we come out well from the comparison. In other words, we are motivated by Tosi and Warmke's *recognition desire* ("we seek beneficial comparisons that attest to our worth because we want to be admired and respected for our achievements" (Kaftanski 2021, p. 210)). Key texts in which Kierkegaard warns against the dangers of comparison include "Our Duty to Remain in Love's Debt to One Another" in *Works of Love* and several of the lily and bird discourses. In the former, he argues that comparison is "the most disastrous association that love can enter into" and "the worst of all seductions" (WL 186/SKS 9, 186), offering a series of images (swampy ground, a hidden worm) as to how the spirit of comparison destroys love. His advice is that we should focus on the tasks of love, using an image reminiscent of the tax collector's standing "far off": Luke 10:4's advice to "greet no-one along the way" to avoid being seduced by comparison. The implication for the reader familiar with the biblical text is that to be tempted by comparison is to be like a lamb sent out amongst wolves (Luke 10: 3).

The parable of the "worried lily" focuses on how comparison exacerbates worry: by comparing ourselves to others ("Where do I rank?"), we over-emphasise or exaggerate the diversity between us rather than resting content with our common humanity (UDVS 159–182/SKS 8, 259–80), a fundamental aspect of Kierkegaard's Christian worldview. Kaftanski argues that social comparison is oriented towards imitating the desires of others, *à la* René Girard (Kaftanski 2021, p. 213). On this reading of the relevant lily and bird discourse, the bird encourages the lily to desire what the bird desires. This does not seem quite right: it would be more accurate to say that—as in advertising—the bird (once again described as a "seducer" (UDVS 169/SKS 8, 269)) persuades the lily to desire something *she had not previously desired or been aware of desiring* (to flourish elsewhere, where she might somehow become the most gorgeous of lilies, the Crown Imperial). Operating here is the "busy or worried inventiveness" and the "restless mentality" of comparison (UDVS 165, 169/SKS 8, 265, 269). When this is in play, the "reference networks" on which defenders of virtue-signaling rely can lead us astray (as with the lily, whose agreement to being uprooted to join the Crown Imperials leads to her death). The more general point for our purposes is

that our motivations (including the "recognition desire") are often generated by desires that are *rooted in vices*: most obviously, pride (in its various forms) or envy. This can be true of both individuals and groups. Comparison confuses us as to what our genuine ethical and spiritual needs are, simultaneously undermining our authenticity or deep character. This brings us to a consideration of character, genuine virtues, and their relation to Kierkegaard's notion of the "single individual".

*4.2. The "Single Individual" and the Virtues*

The "single individual"—the person responsible before God (cf. PV 111/SKS 16, 91)—is one of the most famous phrases associated with Kierkegaard. Yet, the connection between this concept and the virtues has rarely been explored. An exception is found in a recent book by Robert C. Roberts. Roberts argues that being a "single individual"—like other Kierkegaardian terms, such as "earnestness" (*Alvor*) and "purity of heart"—is not a virtue as, say, courage or patience is but rather "a characteristic without which a person cannot be properly or deeply courageous or patient or exemplify any of the other virtues" (Roberts 2022, p. 16). Being a single individual is at the heart of what it means to have "character". Being a single individual, in other words, offers "a kind of moral solidity or constancy in the face of social pressures" (Roberts 2022, p. 17). "Constancy", in turn, is that without which, according to Alasdair MacIntyre, "all the other virtues to some degree lose their point" (MacIntyre 1985, p. 242). Unless I have such constancy—a kind of inner integrity—I cannot manifest a quality of character over time and over different situations: that is, manifest lasting reliable traits in the way characteristic of possessing a genuine virtue.

Consider how the virtue-signaler falls short when viewed through this lens. Westra's groups are different in their character and motivations. The "sincere advocates for change" are *passionately* committed to the desired new norm. The virtue-signalers, by contrast, being primarily motivated by social comparison and the recognition desire, ask themselves, "What do most people in this group think, and how do I gain credit by showing that I am on their side?" I shall argue that there is a family resemblance between the virtue-signaler and the attitudes Kierkegaard associates in *A Literary Review* with the "present age" of "reflection" that he contrasts with the "age of revolution". The central contrast is roughly this: if "passion" is what drives a life lived according to ideals that make normative demands on us, then "reflection" (in Kierkegaard's pejorative sense) is how we talk ourselves into evading those demands—in part through a "shrewdness" to which we shall return later. But to see the connection with virtue-signaling, consider first *A Literary Review*'s headline claim about the present age, namely that it is "essentially a *sensible, reflecting age, devoid of passion, flaring up in superficial, short-lived enthusiasm and shrewdly* [klogtigt] *relaxing in indolence*" (TA 68/SKS 8, 66, emphasis in original).[29] Call this "the present age passage".

This passage (key elements of which I aim to unpack later) may usefully be contrasted with the following slightly later claim: "Morality is character; character is something engraved;[30] but the sea has no character, nor does the sand, nor abstract common sense, either, for character is inwardness" (TA 77-8/SKS 8, 75). The sea has no character in the sense of having no *definite form*, assuming the shape of whatever contains it (e.g., harbours) or the external forces that act on it (e.g., gale-force winds). The kind of person described in the "present age passage" is similarly shaped by the social circle in which they find themselves. Allowing oneself to be shaped by a given group is explicitly what virtue-signalers do. But such people are precisely those who we might describe as "lacking moral fiber" (Roberts 2022, p. 54) in the sense that without lasting, reliable traits, such a person lacks the inner structure to hold their "shape" in the face of difficulties, blown about by the changing winds of fads and fashions. In this way, virtue-signalers *lack character*, which for Kierkegaard is both a moral and a religious failing. If part of the lesson of the worried lily parable is not to act against one's nature—not to try to be what one is not—then the problem with the virtue-signaler is that there is nothing, in this sense, that they are.

But surely, someone might object, we human beings are necessarily influenced by each other, and such influence is part of our moral and religious education and growth? Kierkegaard, after all, has much to say about the importance of ethical and religious exemplars (Socrates, Christ as Pattern as well as Redeemer) and the value of "imitation" (which in *Practice in Christianity* he contrasts favourably with mere "admiration" (PC 237-57/SKS 12, 230–49)). Thus, his position can hardly be to commend a norm according to which we are all utterly isolated from each other.

This objection is on to something. So, to better understand more precisely what Kierkegaard's worry is, we should turn to his concerns about mass society. His primary worry is about a particular kind of "reference network", "the crowd" or "the public", which is an obstacle to the discovery of ethical-religious truth. However, I shall argue, his concerns about the dangers of such collectives—such as the threat posed by such phenomena as "leveling"—would extend to any reference network in which the need to develop "character" as a "single individual" is downplayed. This, we shall see, gives us additional reasons to reject defences of virtue-signaling.

*4.3. The Danger of "the Public"*

Kierkegaard was famously sceptical about the arrival of mass society, seeing "the crowd" as a threat to the single individual. "The Crowd is Untruth" is a well-known Kierkegaardian slogan,[31] and his analysis is widely recognised as anticipating many later analyses of crowd psychology. While much work has shown that the caricature of Kierkegaard as a straightforwardly anti-social or asocial thinker is greatly overblown (see, e.g., Pattison and Shakespeare 1998; Lappano 2017), it remains true that there are dangers in human collectives about which he is acutely concerned. Kierkegaard's account of various such collective entities ("the crowd", "the public", "humanity *in abstracto*" (SUD 31/SKS 11, 147)) tends not to distinguish between these overlapping notions in any systematic way. That said, one distinction between the crowd and the public noted by Kaftanski is useful for our purposes. Whereas "crowd" suggests a gathering of people, often physical, in a certain place (think of a football crowd), "the public" is typically formed "around a particular idea or value and does not need physical proximity to generate the power of influence" (Kaftanski 2021, p. 188). Kaftanski gives as an example the readership of a newspaper,[32] but a more contemporary example of obvious relevance to our concerns here is a "community" formed on social media. Yet, it has been suggested that even such a virtual public, just like a *physical* mass of people, can threaten "a lack of rational control over one's opinions, credulity, vulnerability to emotional contagion, psychic suggestibility, and, more generally, an inclination for…imitation" (Lawtoo 2013, pp. 104–5, cited in Kaftanski 2021, p. 189). With that in mind, we can ask: precisely what is the problem with "the public"?

A vital part of Kierkegaard's answer is that "the public" is *the instrument through which "leveling" is achieved*. Leveling [*Nivelerringen*] is the process of smoothing out—and thereby smothering—the exceptional, the excellent, the "other". A "lowering of moral and intellectual expectations fortified by unified mutual reassurance" (Kaftanski 2021, p. 187)— a decent description of the epistemic over-confidence caused by an epistemic bubble or echo chamber—tends to erode excellence (compare here our earlier concern about moral regress).[33] Whereas "approximate" leveling can take place by, say, the union of essentially weaker forces to neutralise a stronger force (TA 90/SKS 8, 86), Kierkegaard claims that for leveling proper to take place, what is needed is a "phantom", the very "spirit" of leveling, and that phantom is "the public", which he describes as "a monstrous abstraction, an all-encompassing something that is nothing, a mirage" (TA 90/SKS 8, 86). The appearance that it gives of being a *genuine group* is a sham. This phantom is created by a toxic combination of the "passionlessness and reflectiveness of the age" and "the press" (TA 93/SKS 8, 89). The point here is that "the public" is a bogus entity, being claimed to be a whole—a genuine group—yet really consisting of "unsubstantial individuals who are never united or never can be united in the simultaneity of any situation or organisation and yet are claimed to

be a whole. "The public is a corps, outnumbering all the people together, but this corps can never be called up for inspection; indeed, it cannot even have so much as a single representative, because it is itself an abstraction" (TA 91/SKS 8, 87). Yet, this "phantom" can be pressed into service as that on behalf of which any number of deeds are said to be prosecuted, hiding behind such claims as "public opinion demands" this or "the will of the people" is that.

How does this phantom bring about "leveling", and what problems arise from leveling?

### 4.4. Tall Poppy Syndrome

Kierkegaard claims that the vice of *envy* is the principle that (negatively) unifies a "passionless and reflective" age (TA 81/SKS 8, 78). Envy—which Kierkegaard sees behind much praise for equality—is the motive for the mutual reassurance of lowered expectations mentioned above. He distinguishes between *ethical* and *characterless* envy. The problem from which the former arises is the inability, rooted in human nature, continually to admire the truly excellent. The solution is to joke enviously about such excellence, and there is no huge problem with this, provided such joking provides merely temporary relief, thereafter returning the participants to a proper recognition of excellence. Kierkegaard suggests that the practice of ostracism in ancient Greece was an expression of such an "unhappy infatuation" of envy—but in such a way as to remain a genuine recognition of excellence. His example, from Plutarch, is the man who confessed to the Athenian statesman Aristides that he was voting to banish him due to weariness at hearing Aristides everywhere called the only just man—a confession that, Kierkegaard notes, reveals something about the man, while doing nothing to diminish Aristides' excellence (TA 83/SKS 8, 80).

However, a crucial shift takes place when "reflection becomes dominant and develops indolence" (TA 83/SKS 8, 80). At that point, envy becomes "characterless" and unaware of its own significance. Unlike the banish-Aristides-voter, who (unlike Havel's greengrocer) at least recognises his own low motives, Kierkegaard's concern here is with an envy that—say, in wanting to cut such a "tall poppy" down to size—continually changes its story as to whether its critical comment was a joke, an insult, "ethical satire", or nothing at all (TA 83/SKS 8, 80) based on the reactions it receives. It attacks its target but—on receiving a more negative reaction than expected—hides behind the claim that this was nothing at all ("Only joking!").

Leveling, then, is how Kierkegaard claims the vice of envy manifests itself (TA 84/SKS 8, 80). However, I suggest that Section 3 above has given us reasons to see how another vice in which the comparative dimension is central—self-righteousness—can also inspire leveling (recall especially the discussion of silencing). Either way, such leveling gives rise to various problems. First, since by leveling's very nature no particular individual can take the lead, leveling is "abstraction's victory over individuals" (TA 84/SKS 8, 81). Its power is such that no individual will be able to halt it (TA 87/SKS 8, 83). The values of an age in which leveling is dominant are formed by consensus, but the problem with this is that this is not a genuine consensus but the "victory over individuals" *of "the public"*. A consensus that leaves out individuals is inauthentic (cf. Kaftanski 2021, p. 189) because it lacks the crucial element of individual responsibility: as Kierkegaard puts it elsewhere, the crowd "weakens responsibility by reducing the responsibility to a fraction" (PV 107/SKS 16, 88). And since the "individual" alone before God is a fundamentally religious category for Kierkegaard, the loss of the individual is a religious catastrophe.

Second, as our discussion of envy has already suggested, those who challenge the lowered bar of social norms are likely to be ridiculed and rejected (cf. the ostracism discussion and, as the opposite side of the coin, our earlier discussion of silencing). And third, it is not just brave outsiders who are damaged. Consider the following passage:

> Leveling is not the action of one individual but a reflection-game in the hand of an abstract power. . . . [T]he individual who levels others is himself carried along, and so on. While the individual egoistically thinks he knows what he is doing, it must be said that they all know not what they do, for just as inspired

> enthusiastic unanimity results in something more that is not the individuals', a something more emerges here also. A demon that no individual can control is conjured up, and although the individual selfishly enjoys the abstraction during the brief moment of pleasure in the leveling, he is also underwriting his own downfall. (TA 86/SKS 8, 82-3)

In other words, even enthusiastic participants in leveling are in danger, since if you give the public an inch it will take a mile. Any individual who participates "underwrites his own downfall" in the sense that the moral, social, and religious damage ultimately caused by leveling rebounds even on those who think they may gain from it. Kierkegaard's solution to all this is the "infinite liberation of the religious life" (TA 85/SKS 8, 82), which genuinely recognises individual uniqueness (TA 87-8/SKS 8, 83-4).

Consider the indented passage above in connection to virtue-signaling, and a contrast between "enthusiasts" and "levelers", roughly paralleling Westra's "sincere advocates for change" and his virtue-signalers. If Kierkegaard is right, then both the original advocates and the virtue-signalers are participating in a process the results of which cannot be controlled. There are at least two risks here: that the change is moral regress rather than progress; and that even if, in at least one sense, the change is positive, plenty of (potentially negative) unintended consequences can arise. Of course, this is not an argument always to favour the status quo! But it does mean—and this is my primary point—that at the heart of Westra's rejection of the idea that virtue-signaling is a social ill (Westra 2021, p. 157) is a failure to recognise the dangers of leveling and how social change can involve these dangers just as it can involve genuine progress. Moreover, those who, like Kierkegaard, are suspicious of the collective actions of the inauthentic might worry not only that (as suggested earlier) Westra has begged the question in favour of moral progress in the examples that he gives in support of virtue-signaling. They might also that when "the public", "the crowd", or, today, the online lynch mob is in the driving seat, the likelihood of moral *regress* may outweigh the likelihood of moral *progress*. At the very least, this is a serious possibility that should not be overlooked.

This highlights a key question that defenders of virtue-signaling tend to ignore, namely: *Can we afford to overlook the importance of character?* A person's ethical or religious character cannot be divorced from their passions: what they care about. The virtue-signaler cares about how they will be perceived by a particular in-group and is thus ultimately dependent upon what some analogue of "the public" thinks and cares about. Kierkegaard's "worried lily" parable teaches us that this is often rooted in a worry inspired by comparison. As Roberts puts it, the crowd or public is "a social force of public opinion, a system of anonymous felt approvals and disapprovals that pressure people without character to conform and leaves them without a stable and deep moral anchor" (Roberts 2022, p. 56).[34] In *The Sickness Unto Death*, Kierkegaard claims that the "criterion for the self is always: that directly before which it is a self" (SUD 79/SKS 11, 193). A cattleman who is a self directly before his cattle, or a master before his slaves, lacks an appropriate criterion or yardstick (*Maalestok*). Yet, if the virtue-signaler's criterion is some version of "the public", one of the key features of the public is its fickleness. In a famous image, Kierkegaard muses: "If I were to imagine this public as a person . . . I most likely would think of one of the Roman emperors, an imposing, well-fed figure suffering from boredom and therefore craving only the sensate titillation of laughter . . . So this person, more sluggish than he is evil, but negatively domineering, saunters around looking for variety" (TA 94/SKS 8, 90). The whims of such a collective Nero-figure are no yardstick by which to guide one's life.

By contrast, "the single individual"—understood as someone with character—can stand fast against the collective pressures that influence the morally shapeless. One way of thinking about the quality of being a single individual is that it gives the power to resist a particular kind of "environmental vicissitude" (Roberts 2022, p. 56): the crowd or public.

A possible objection here is that virtue-signalers respond to real groups, not phantoms. But this distinction cannot so easily be drawn: as we saw in Section 2, social media environments make it hard to be confident about the size or significance of the group. One

can be confident that *some* people hold a particular view (just as the views attributed to "the public" are held by *somebody*), but one cannot be confident about how many, still less that this is where genuine "public opinion" sits.

### 4.5. "Shrewdness" and Its Relevance to Virtue-Signaling

Especially notable in the "present age passage" is that quality of which Kierkegaard is suspicious, *Klogskab*, which we may think of as in one sense the opposite of "passion" (as we saw, Kierkegaard in that passage describes the age as "devoid of passion"). Though occasionally used in a more neutral sense,[35] *Klogskab*—typically translated by the Hongs as "sagacity" but the meaning of which would be better captured by "shrewdness"—is normally used by Kierkegaard in a pejorative sense. There is something *calculating* about it—typically, calculation that one believes to be in one's self-interest, echoing a phrase Kierkegaard uses to describe the opposite of inwardness: being "a spectator computing the problem" (TA 79/SKS 8, 76). (In the "present age passage", I take "sensible" to mean "calculating" in this sense.) Even defenders of virtue-signaling such as Westra acknowledge the degree of self-interested calculation[36] that it involves. There is a clear difference here between the attitude of the virtue-signalers and the kind of passionate commitment of the "sincere advocates for change". Virtue-signalers, lacking steadfastness or "constancy", are more likely to be fair-weather friends of the changes in question, echoing Kierkegaard's thought that "there is always a connection between flashes of enthusiasm and prudential apathy" (TA 73/SKS 8, 70-1). This latter phrase also describes the nature of many social media pile-ons. The online lynch mob can be ruthless in its attack on the latest victim, before quickly tiring of this target and moving on (such pile-ons thereby echoing the "superficial, short-lived enthusiasm(s)" (*Begeistring*) of the "present age passage"). Nevertheless, the ruthlessness that precedes the Nero-like boredom is part of the dark side of piling on: it can be a form of bullying that violates any reasonable principle of proportionality.[37] It reminds one of what Kierkegaard elsewhere calls the "cannibalistic taste for human sacrifices" (CD 340/SKS 14, 94) and illustrates his claim that not even the most cowardly individual "was ever as cowardly as the crowd always is" (PV 108/SKS 16, 88).

However, given Kierkegaard's astute awareness of the human capacity for self-deception, I think it makes sense to see in the "present age passage" two distinct character flaws: one focused on this tendency towards "superficial, short-lived enthusiasm" (often linked to self-deception) and the other on its allegedly "sensible" lack of passion (which also explains its ability "shrewdly" (*klogtigt*) to "relax in indolence"). These two foci suggest two different kinds of virtue-signaler, whom I shall label *superficial enthusiasts* and *clear-eyed cynics*.

*Superficial enthusiasts* lack the ability to track their own unreliable emotions. Unable through self-deception to recognise the work that their recognition desire plays in their motivations, such people take themselves to have character traits they really lack (for instance, mistaking as a deep concern for social justice or love of their country their own need to fit in with a given in-group). Lacking the constancy of "character", whole groups of such people can be influenced to adopt certain values by sufficiently charismatic people with social capital. If Westra's "sincere advocates for change" include amongst their number these charismatic influencers, then they have significant power to influence such enthusiasts. Recalling Havel, such a greengrocer then puts up his sign not *in order* to be in harmony with society but because he *is* in harmony with society (Forsberg 2021, p. 73). In ways we shall shortly see, superficial enthusiasts then end up running the same vice-risk—self-righteousness—as the sincere advocates for change, as the result of social contagion. However, self-deceived as they are about their own motives, what superficial enthusiasts take to be a genuine commitment may only be "fair-weather". Their enthusiasm is likely to be short-lived, prone to being displaced by the next fashionable cause.

*Clear-eyed cynics* are more straightforwardly "shrewd", as opposed to self-deceived. (Thus Groucho Marx: "Those are my principles; and if you don't like them, well, I have others.") They self-interestedly calculate which causes they should publicly support based

on what will win them the most praise or acceptance with the relevant group. (So, a greengrocer who puts up his sign because he *wants* to appear to be in harmony with society.) In this way, the clear-eyed cynic is an example of using reflection (*Reflexion*) and shrewd deliberation in a "prudent", passionless manner, a key theme in *A Literary Review*. Such a person is motivated by what Kierkegaard elsewhere calls "probability", which is to fall short of what is demanded of us religiously, since "all religious venturing, to say nothing of the Christian . . . is by way of relinquishing probability" (JFY 100/SKS 16, 157). And sometimes, these cynical calculations are anyway based on highly limited horizons. Roberts offers the image of reflection as a spider's web, noting that like flies born in such a web, many people are unaware that they are caught in the web of problematic reflection, misconstruing their ability to flutter their wings slightly as genuine freedom (Roberts 2022, p. 167). In a similar way, many a clear-eyed cynic makes their choices from a needlessly limited menu. As we noted earlier, this can lead to their "downfall" in the sense that they aid and abet the spread of damaging ideas—to the disadvantage of the society of which they are part.

The existence of clear-eyed cynics underlines another point against Levy's defence of virtue-signaling. They enable us to see that the confidence he claims virtue-signalers can have about norms (based on the strength of emotion and the weight of numbers) is not well-founded confidence. The strong emotions clear-eyed cynics express are not genuine: they are expressed to impress or to "pass". When trumping up or excessive outrage, for instance, involves faked, inauthentic responses, these are not reliable signals. Moreover, as Hill and Fanciullo point out, virtue-signaling standardly takes place in an environment in which group identity is vital, and once a view becomes established as dominant within that group ("This is what the public thinks/demands!"), many virtue-signalers will pile on to align themselves with the dominant view (perhaps motivated by moral cowardice). It would be mistaken to count this as additional evidence in favour of the dominant view: to do so amounts to a kind of double-counting independent of the merit of the view in question (Hill and Fanciullo 2023, pp. 117, 16).

But superficial enthusiasts and clear-eyed cynics have the following in common. By appealing to some version of "the public" as authority, both can spread the impact of the vice of self-righteousness, though in different ways. The first kind of virtue-signaler, lacking the ability to spot their own unreliable emotions, can gain illegitimate, bootstrapped confidence about the judgements they aim to support. For this reason, they end up running the same vice-risk—self-righteousness—faced by the original advocates for change. The second kind of virtue-signaler, while lacking the passion typical of the self-righteous, nevertheless aids the spread of that vice in the manner of the calculating bystander. It is not that they succumb to self-righteousness through social contagion; rather, they are not bothered by the damage caused by self-righteousness (as outlined in Section 3), provided they themselves benefit—or think they will—from the kind of recognition they desire.[38]

In short, Kierkegaard's critique of "the public" shows why we should be wary of the insufficiently critical reliance on social norms made by virtue-signalers and their defenders. His analysis of the deficiencies of the "present age" and its contrast with genuine "character" offer reasons not to rest content with the lack of deep character of the virtue-signaler. Deference to social norms is no substitute for the inner integrity Kierkegaard associates with the person of character.

## 5. Summary and Conclusions

This article has argued that the most prominent defences of virtue-signaling fail. Levy's defence based on higher-order evidence and the weight of numbers underestimates the risks of epistemic bubbles, echo chambers, and the problems of "double-counting" that make this evidence and these numbers unreliable. Westra's defence based on virtue-signaling's potential to contribute to moral progress begs the question, as equivalent methods can contribute to moral regress. In considering the vice-risks in the arena of virtue-signaling, I have argued that self-righteousness is an underexplored vice that can do moral, social,

and epistemic damage (complementing Kierkegaard's discussion of envy). Kierkegaard's contrast between the person whose reference networks effectively appeal to the authority of collectives, such as "the public" or their analogues, and the genuine person of character deepens our understanding of why we should reject defences of virtue-signaling. "The public" and its correlates are an inadequate yardstick for our norms. It is no substitute for the development of moral and religious integrity, constancy, and genuine "character". Moreover, this analysis suggests the value of distinguishing the kind of vice-risks run by Westra's "sincere advocates for change" (self-righteousness) and their virtue-signaling followers (lack of character). However, even the latter can spread the moral, religious, social and epistemic harms of self-righteousness, by their failure to care about these risks, which they either overlook (superficial enthusiasts) or to which they give less weight than the recognition desire (clear-eyed cynics). By introducing Kierkegaard into the virtue-signaling debate, I have sought to show more clearly why, *pace* Westra, there are good reasons to be worried about the potential of virtue-signaling to contribute to moral regress, rather than simply celebrating its potential in moral progress. *Pace* Levy, virtue-signaling is *not* virtuous. It is, typically, far closer to vice than to virtue.

**Funding:** This research received no external funding.

**Acknowledgments:** I am grateful to C. Stephen Evans, Timothy Smartt, various participants at the School for Virtue and Character conference at the University of Notre Dame Australia, and three referees for this journal for comments on earlier versions of this article.

**Conflicts of Interest:** The author declares no conflict of interest.

## Notes

[1]   In line with the bulk of the literature, I take moral grandstanding to be a roughly equivalent term to virtue-signaling. For reasons to be given shortly, I shall treat the two terms interchangeably.

[2]   The British journalist who claims to have coined the phrase characterises virtue-signaling as "indicating that you are kind, decent and virtuous", holding an "approved, liberal media-elite" set of views in such a way as not to demand any costly action (Bartholomew 2015).

[3]   See Walsh (2018) for a highly sceptical view of this connection and Tietjen (2013, esp. pp. 117–34) and Roberts (2019, 2022) for far more positive views. For my own position on this issue—far closer to Roberts than Walsh—see Lippitt (2020, esp. pp. 162–6).

[4]   For more on each of these points, see Lippitt (2020, pp. 162–6).

[5]   Levy and Westra both treat grandstanding and virtue-signaling as interchangeable terms (Levy 2021, p. 9545; Westra 2021, p. 157n3). While giving reasons why they prefer the former (Tosi and Warmke 2020a, pp. 37–40), Tosi and Warmke acknowledge that "[w]hat many people mean by 'virtue signaling' is often very close to what we have called 'moral grandstanding'" (Tosi and Warmke 2020b). In following this usage, I shall not be using the term "virtue-signaling" in so expanded a way as to include every instance of a social media utterance that might be labelled as such by a hostile reader.

[6]   All references to Kierkegaard's texts (Kierkegaard 1967–1978, 1978, 1980, 1990a, 1990b, 1991, 1993, 1995, 1997, 1997–2013, 1998, 2011) will be made parenthetically as above: first to the appropriate English translation, using a widely used sigla, as noted in the References section, and second to the now standard Danish edition (also listed in the References section), designated as SKS, followed by the volume and page number.

[7]   Although Tosi and Warmke and their critics discuss grandstanding in a predominantly secular context, it will be important for our purposes to bear in mind that the phenomena they describe sometimes take place with respect to religious in-groups. Throughout the following discussion, therefore, I invite the reader to think of the term "moral" in a broad sense that can include religious commitments, including, for instance, the expression of certain views on, say, abortion, euthanasia, or warfare, on religious grounds.

[8]   They combine both kinds of grandstanding under the broad heading of "respectability". But the distinction between the desire to pass as morally decent and the desire to appear unusually morally impressive (or "holier than thou") is significant.

[9]   This paper's empirical work suggests that the motivation for moral grandstanding is "associated with status-seeking personality traits" (Grubbs et al. 2019, p. 1).

[10]   For a detailed analysis of proper and improper forms of self-love in Kierkegaard, see Lippitt (2013).

[11]   To anticipate a point we shall return to in Section 4, this pressure may be felt with greater or lesser degrees of conscious awareness.

[12]   Hence "moral outrage porn", in which representations of moral outrage are engaged with "primarily for the sake of the resulting gratification, freed from the usual costs and consequences" of engaging with such content (Nguyen and Williams 2020, p. 148).

13  The focus on effects does not ipso facto favour a consequentialist approach: they also critique grandstanding based on failures of *respect* and on what it says about the *character* of the grandstander (Tosi and Warmke 2020a, pp. 97–137).

14  Recall here that I am using the term "moral" in a broad sense to include certain forms of religious commitments—so in such cases, we could read "reliable religious cooperators".

15  The same distinction also undermines the second claim about signaling who are reliable moral co-operators: all that is required here is norm-signaling, not virtue-signaling (Hill and Fanciullo 2023, pp. 117, 9).

16  To be fair, their original paper does give that impression (Tosi and Warmke 2016, p. 212). In later work, however, they expressly deny that this is their view (Tosi and Warmke 2020a, pp. 72–73). Recall that the primary objection is not to "extreme" views per se but to the process by which group members arrive at them, taken to be epistemically objectionable because of being arrived at by social comparison rather than evidence and argument.

17  Compare, for instance, MacIntyre's (1985) important distinction between goods internal and external to a practice.

18  Hill and Fanciullo also point out that virtue-signalers—unlike direction-enquirers, who are dispassionately truth-directed—have, by definition, ulterior motives: the recognition desire. Such a motive is an unsound epistemic principle on which to act (Hill and Fanciullo 2023, pp. 117, 15).

19  This results from "double-counting" of non-independent judges (Hill and Fanciullo 2023, pp. 117, 16)—more of which is in Section 4.

20  See Nguyen's example of Derek Black, raised by a neo-Nazi father (Nguyen 2020, p. 158).

21  The person raised in an echo chamber is plausibly described as a victim of "hermeneutical injustice" (Fricker 2007, pp. 147–76). Their powerlessness prevents them from generating the conceptual resources necessary properly to understand their social experience—akin to a woman experiencing sexual harassment before there was any such widely understood concept. They are "hermeneutically marginalized" (p. 153) in that they are excluded from a practice that would benefit them: open discussion in a context where their worldview has not been skewed by the manipulation of their trust.

22  Westra makes *some* attempt to address problems of bias arising from polarisation (Westra 2021, pp. 173–5) but without reference to epistemic bubbles and echo chambers and in a way that underestimates the seriousness of the problem.

23  Indeed, Westra practically invites the epistemic bubble problem when he acknowledges that a person's "reference network" is "the group of people whose behaviors and attitudes they care about" (Westra 2021, p. 160).

24  Even in such cases, it would still be objectionable qua "vanity project" and—to anticipate Section 4—at the level of the "character" of the virtue-signaler.

25  This echoes Kierkegaard's language of "going out" amongst the lilies "to avoid all comparison with human beings" (UDVS 167/SKS 8, 266).

26  I shall follow Rorty in talking of "morality", but again, I shall be taking the term to include moral judgements rooted in religious convictions. Plausibly, the same problems may face *specifically* religious judgements in a parallel way.

27  This is a particular problem for the epistemically underconfident (see Plakias 2020) and may be especially marked in religious organisations with hierarchical power structures.

28  Tanesini and Goldberg are discussing intellectual arrogance, not self-righteousness. There is a significant overlap between these two concepts, though their precise relationship is not my concern here. While self-righteousness often expresses arrogance, it is not straightforwardly a sub-category thereof: one can be self-righteous in matters where one does not think of oneself as intellectually superior but rather assumes that *any* decent person would think as one does—just not those appalling people on whom one sits in judgement. (Recall the two different desires Tosi and Warmke treat under the "respectability" heading.) However, because of the overlap, I assume in what follows that in many relevant cases, claims about intellectual arrogance also apply to self-righteousness.

29  I have altered the translation of *klogtigt* from "prudentially" to "shrewdly". Typically, the Hongs translate *Klogskab* as "sagacity". I shall explain shortly why I prefer "shrewdly"/"shrewdness".

30  Kierkegaard here trades on the etymological origins of "character", which can be traced back to the Greek *charassein*, meaning to sharpen, make pointed, or engrave. The character in which he is primarily interested is *Christian* character.

31  See especially the first of two notes on "The Single Individual", described as notes on "my work as an author", in which "the single individual" is repeatedly contrasted with "the crowd" and the latter rejected as a source of ethical-religious truth (PV 105-112/ SKS 16, 85–92).

32  This is unsurprising: the obvious reference is to the notorious "*Corsair* Affair", in which Kierkegaard was ridiculed by a satirical publication in a way that significantly impacted his daily life.

33  Kaftanski explicitly refers to the "echo chamber effect" (Kaftanski 2021, p. 187)—though he does not analyse echo chambers or epistemic bubbles.

34  Once again, we need to read "moral" here in a broad sense that includes the religious.

35  For instance, in one of the 1844 upbuilding discourses, Kierkegaard describes virtue as "the highest sagacity" (EUD 380/SKS 5, 363), contrasting this with the idea of sagacity held by a "sensate person". The person holding the former conception of sagacity

recognises virtue as the optimal way to live, but the sensate person will not see this unless their entire conception of what is really in their best interests changes. For further discussion, see Roberts (2022, pp. 102–3).

36  "Self-interested", that is, in something more like the sense held by the "sensate person" than that held by the virtuous person, focusing on an anticipated worldly advantage.

37  Ronson (2015) gives a vivid account of how the lives of victims of public shaming arising from Twitter storms can be left in tatters after the mob has moved on.

38  In the first note on "The Single Individual", Kierkegaard puts it like this: "To love the crowd, or pretend to love it, to make it the authority for *the truth*, is the way to acquire tangible power, the way to all kinds of temporal and worldly advantage—it is also untruth, since the crowd is untruth" (PV 111/SKS 16, 97).

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
