# Peer review of "Kierkegaard, “the Public”, and the Vices of Virtue-Signaling: The Dangers of Social Comparison"

_religions, doi:10.3390/rel14111370_

Round 1

Reviewer 1 Report

Comments and Suggestions for Authors

Review of Kierkegaard, “the public” and the vices of virtue-signaling for Religions

Strongly recommended for publication. This review signals minor typos and food for thought for the author(s). Some observations and recommendations below.

Overall. Strong paper with a clear set of goals and meticulous argumentation to their ends. The paper not only offers a persuasive criticism of virtue signalling/grandstanding in the contemporary literature using Kierkegaard’s thought, but also makes Kierkegaard’s philosophy relevant to key contemporary social-moral-religious (and political) issues.

1.     The paper is richer than the title suggests. By incorporating “social comparison” or “comparison” in the title (term(s) crucial to the paper’s argument) this paper could end up on a radar of a larger audience that would benefit from its wisdom (non-Kierkegaardians, non-philosophers, beh. & soc. psychologists, etc.)

2.     The author contributes a useful distinction of 2 types of virtue-signallers – perhaps it would be important to include superficial enthusiasts and clear-eyed cynic in the abstract.

3.     The paper gives a concise account of virtu-signalling/moral grandstanding, focusing on relevant exemplary literature, including criticisms, plus their own strong argumentation that is both synthetic and novel.

4.     (minor) While it is clear that the author is well-read in the relevant literature, it seems that this sentence below represents a rather large basket of characteristics of SK’s position on virtues. Perhaps breaking it down and providing more references would be helpful for readers.

It is usually agreed that if Kierkegaard is such, he is one of a very specific kind, one who:  aims at the “upbuilding” of the whole person, focusing on their religious and not just  moral development; whose approach to the virtues therefore incorporates specifically religious conceptions of faith, hope, love, gratitude, etc.; who denies that the cultivation of virtues is in any way “meritorious”, the development of such qualities being a gift of grace; and who (perhaps implicit in all of the above) is especially concerned with how virtues are transformed by Christian revelation and faith.

5.     In the sentence above the phrase “in any way meritorious’”  seems like a strong claim (considering the relevant literature, see: Davenport), perhaps it needs rephrasing.

6.     All honour to such scholarship, as Johannes Climacus would say.

Perhaps adding “Kierkegaard’s pseudonym” would help linking JC with SK for a non-specialist audience. (53)

7.     Perhaps adding an additional sentence to the sentence below would further clarify the ambitions of the author. It would tie well with the author’s following sentence that starts at (57)

·      But my concern here is different: to connect Kierkegaard 54 for the first time with the philosophical debate about the qualities – virtuous or vicious – 55 of virtue-signaling, and to show what his thought can contribute to this debate.

·      I show how SK’s view of x helps us understand that y is bad because it is vicious.

8.     “illegitimately inflated” (61)– little unclear/opaque

9.     Hard to read: “If alternative views are actively discredited by trust- 363 manipulation, then what are ordinarily epistemically virtuous activities can lead to the 364 biases of the echo chamber being reinforced.

10.  There seems to be some kind of flattening and perhaps a reduction/a contraction of 2 problems into 1 (“be wary of the insufficiently critical reliance on social norms made by virtue-signaling’s defenders”) in the paper overall. One thing is to be wary of the insufficiently critical reliance on social norms (which is at times argued as being problematic as such by drawing on various examples and justifications such as problems with moral progress etc.) and the other thing is that there is something specifically wrong in how virtue signalling defenders contribute to that problematic phenomenon (a view justified on dispositional [character] grounds). Perhaps it would be useful to distinguish these two.

11.  Does the Pharisee example meet the second criterion of grandstanding? (The “grandstanding expression”, which can be written or verbal, 106 amounts to aiming to satisfy that desire via public moral discourse)? Seems like there is a problem with that on p. 12 where it seems this will be argued (especially 481-486 passage that seems to be explaining the link between grandstanding and the example from NT). I am unable to see how self-righteousness is meant to lead to public (written/verbal) expression in the case of the Pharisee. Perhaps I am reading too much into it.

12.  There is a bit of a confusion in the indication of particular sections and the line of arguments ((481)first, (487)second, (494)first.

13.  (552) “ ‘“images”’ – “mental” images?

14.  Having in mind the “To Kill a Mockingbird” example, could we say the same thing in the sentences below using the words “bias” or “biased” rather than “self-righteousness”? Do self-righteousness and bias have the same role here? Different? One instigates the other? One is a subcategory of the other?

We can therefore see that self-righteousness [BIAS] has the capacity to inflict significant epistemic harms on those over whom the self-righteous [BIASED]stand in judgement – especially if 566 the self-righteous [BIASED]have the weight of in-group numbers on their side. Being judged in this 567 way can lead, first, to being silenced, and a consequent erosion of self-confidence which 568 over time can lead to what Alessandra Tanesini calls “servility”:

15.  Typo (642) “brought up I such a manner” – it should be brought up in such a manner”

16.  typo (671-672) “Kaftanksi” should be “Kaftanski”

17.  (706) – perhaps it would be advisable to quote here contemporary literature on this matter that talks about non-fragmentary selfhood, commitment, moral vision, etc. and cross-situational consistency.

18.  (941) Harry Frankfurt talks about a similar issue: what we (in fact) believe in and value and what we think we do and what we wish we did are often different. These differences may not be known to many leading to confusion between them.

19.  Bibliography: perhaps an inconsistency in referencing style pertaining to either Roberts (36: 3) or Tosi-Warmke (44-3).

·      Roberts, Robert C. 2019. “Is Kierkegaard a ‘Virtue Ethicist’? Faith and Philosophy 36: 3, 325-42.

·      Tosi, Justin and Brandon Warmke. 2016. “Moral Grandstanding”. Philosophy & Public Affairs 44-3: 197-217.

Reviewer 2 Report

Comments and Suggestions for Authors

I think the author has done a marvelous job of teasing out the true significance of self-righteousness as a vice, for Kierkegaard. The paper does an excellent job of detailing the direct line from self-righteousness (whether personal or communal/social) to leveling, which Kierkegaard would find exceptionally problematic. But it’s weaker on the connection between virtue-signaling and self-righteousness, which throughout the paper seems only a possibility occasioned by virtue-signaling—not a necessary consequence thereof. As a "folk" term, "virtue-signaling" includes not only grandstanding but also something seemingly less significant: simply sharing in an admittedly superficial way those traits one possesses or actions one has performed which, without making the moral judgment explicit, are expected to reflect well on the individual's character. (For example, a colleague of mine regularly posts photographs of himself volunteering in a local shelter for the homeless to Facebook, with no commentary suggesting what he does is virtuous, typically simply noting that "things are going well tonight at the shelter" or some such; nevertheless, I would think this would qualify in the broader, perhaps more ordinary sense in which we use the term "virtue-signaling," while it would not meet the narrower definition of virtue-signaling as moral grandstanding used in the paper.) While this could easily slip into vicious self-righteousness, it's unclear from the paper why we should believe it's already vicious before it does so -- and much less clear why we might believe that virtue-signaling of this sort necessarily or inevitably becomes self-righteousness.

I admit that this is not the way in which the author uses the term in the paper, and there seems to be solid scholarship supporting the author's use of the term as the synonymous with something more self-evidently problematic -- so this is in no way a criticism of the argument being made. This was simply a point of confusion for me, as a reader, and one the author might wish briefly to address so as to prevent similar confusion in other readers. Other than this point of confusion, however, I see no significant problems with the paper at all.

Reviewer 3 Report

Comments and Suggestions for Authors

This was a well-argued, compelling paper that brought Kierkegaard’s views to bear in an interesting, illuminating way on an important topic. I enjoyed reading the paper, and I think that it is fit for publication. 

I do have several questions and comments, though, which I think the paper could benefit from addressing. (Though to be clear, I don't think these comments need to be addressed for the paper to be published.)

One question I was left with—especially in section 2—is whether the dispute between critics and defenders of virtue-signaling is, in large part, terminological. The third and fourth kinds of moral grandstanding that Tosi and Warmke identity—Trumping Up and Excessive Outrage—are defined such that the badness/viciousness/inaptness of grandstanding is part of the very concept. As you note on page 7, defenders of virtue-signaling (like Levy and Westra) resist such attempts to define virtue-signaling in such a way that it’s by definition vicious. You respond that virtue signaling is a species of norm-signaling with “recognition desire” as its differentium. I think this would be worth foregrounding more clearly earlier in the paper. But I also think it would be worth explaining why Hill and Fanciullo think norm-signaling doesn’t necessarily entail an attempt to gain recognition. On its face, it seems to me that signaling essentially involves communicating something to another party, where that communication aims to get uptake in which the other party recognizes what you’ve communicated. What distinguishes the specific kind of recognition virtue signalers desire from this generic recognition desire involved in any kind of signaling adherence to a norm?

Similarly, I think it would be helpful to clarify whether you are defending the claim that virtue-signaling is always vicious—either as a matter of definition, or on substantive ethical grounds—or whether you think it’s just typically vicious. To my mind, you make a strong case for the claim that virtue-signaling inherently has tendencies to corrupt individuals’ character and have bad social effects, and you do an excellent job of diagnosing what these corrupting tendencies are. But I don’t think it shows that these tendencies are always realized or manifested. That is, your argument leaves open that some genuinely virtuous agents engage in virtue-signaling while avoiding the pitfalls you identity. This also bears on your critique of Westra on pp. 10-11. You say it “still seems a weak defence of the practice that it can lead to desirable outcomes provided the moral change is ‘positive’, when precisely the same process can spread negative moral changes.” But if Westra is just arguing for the claim that contrary to its critics, virtue-signaling is virtuous when performed in the service of promoting better social norms, I don’t think that’s a weak defense. Likewise, one way to defend the epistemic function of virtue-signaling is to say that it’s epistemically valuable when it contributes to a truth-conducive epistemic environment, but it’s epistemically disvaluable when it’s not. (Cf. Jennifer Lackey’s defense of echo chambers.)

Such a defense of virtue-signaling would be especially natural if one accepts an externalist account of epistemic justification. 

I also think it may be worth pointing out some of the deeper points of disagreement between you/Kierkegaard and proponents of virtue-signaling. Kierkegaard thinks a) that striving to imitate others/conform to social norms is generally vicious, and b) that we should not care so much about social recognition, since both of these often hinder fulfilling our fundamental ethical task of becoming a single individual before God. But many philosophers (likely including some defenders of virtue-signaling) reject this view and think that conforming to social norms is, on the whole, a good, pro-social human disposition, and likewise think that human flourishing requires—or is at least significantly enhanced by—social recognition. (As Adam Smith famously said in the Theory of Moral Sentiments, we naturally desire both to be lovely and to be loved.) On this view, conforming with norms the “recognition-desire” of which Kierkegaard is critical can both be healthy and natural. While I don’t think you need to spend more time arguing against these views, it may be helpful to more clearly diagnose what may be some of the underlying disagreements. 

Just one comment regarding the structure: as the paper stands, the first half of the paper explains your own responses to Levy and Westra, and the second half explains how Kierkegaard provides the resources to critique virtue-signaling. I wonder whether the paper could be shortened and streamlined if you combine and integrate parts of each section. 
